# Cystatins from the Human Liver Fluke *Opisthorchis viverrini*: Molecular Characterization and Functional Analysis

**DOI:** 10.3390/pathogens12070949

**Published:** 2023-07-18

**Authors:** Amornrat Geadkaew-Krenc, Rudi Grams, Sinee Siricoon, Nanthawat Kosa, Dawid Krenc, Wansika Phadungsil, Pongsakorn Martviset

**Affiliations:** 1Graduate Program in Biomedical Sciences, Faculty of Allied Health Sciences, Thammasat University, Khlong Luang, Pathum Thani 12120, Thailand; rgrams@icloud.com (R.G.); nantha.ko@hotmail.com (N.K.); wansika_tu@hotmail.com (W.P.); 2Thailand Institute of Scientific and Technological Research, Khlong Luang, Pathum Thani 12120, Thailand; sinee@tistr.or.th; 3Chulabhorn International College of Medicine, Thammasat University, Khlong Luang, Pathum Thani 12120, Thailand; dm_krenc@tu.ac.th; 4Department of Preclinical Science, Faculty of Medicine, Thammasat University, Khlong Luang, Pathum Thani 12120, Thailand; pong_m@tu.ac.th

**Keywords:** *Opisthorchis viverrini*, cysteine proteinase inhibitor, cystatins, cathepsin, inhibition, immune reactivity

## Abstract

A high incidence of cholangiocarcinoma (bile duct cancer) has been observed in Thailand. This usually rare cancer has been associated with infection with the human liver fluke, *Opisthorchis viverrini*. Secretions of the parasite that interact with the host are thought to be a major component of its pathogenicity and proteolysis is a key biological activity of the secreted molecules. In this study, we present a molecular analysis of cysteine proteinase inhibitors (cystatins) of *Opisthorchis viverrini*. Six cDNA coding sequences of *Opisthorchis viverrini* cystatins, *Ov*Cys1–6, were cloned from the adult stage of the parasite using RT-PCR. Based on their sequences, *Ov*Cys1 and *Ov*Cys2 are classified as type 1 cystatins, while *Ov*Cys3–6 are classified as type 2 cystatins, with each containing a signal peptide and only one C-terminal disulfide bond. Their C-terminal region sequences are diverse compared with other cystatin members. Cystatins *Ov*Cys1, 3 and 4 were found in crude worm extracts and excretory-secretory (ES) products from the adult parasite using Western blot detection, while the other isoforms were not. Thus, *Ov*Cys1, 3 and 4 were selected for inhibition analysis and immune reactivity with *Opisthorchis viverrini*-infected hamster sera. *Ov*Cys1, 3, and 4 inhibited mammalian cathepsin L more effectively than cathepsin B. The pH range for their full activity was very wide (pH 3–9) and they were heat stable for at least 3 h. Unlike *Fasciola gigantica* cystatins, they showed no immune reactivity with infected hamster sera based on indirect ELISA. Our findings suggest that *Opisthorchis viverrini* cystatins are not major antigenic components in the ES product of this parasite and that other effects of *Opisthorchis viverrini* cystatins should be investigated.

## 1. Introduction

The human liver fluke, *Opisthorchis viverrini*, is classified as a group I carcinogen that causes cholangiocarcinoma (CCA). This cancer type has a high incidence in Thailand and other countries in Southeast Asia [1,2,3]. One proposed mechanism contributing to CCA through chronic infection is the damage caused by components of the excretory-secretory (ES) products of the parasites. The released proteins can cause the induction of cell proliferation and inhibition of DNA repair/apoptosis [3,4]. Proteinases and their inhibitors are major components in the ES products of many parasites and are involved with many biological functions in parasites, including nutrition, immunomodulation and host-tissue digestion [5,6,7]. Consequently, it is necessary to develop strategies that interrupt parasite infection. Proteinase inhibitors are of interest as they not only regulate the proteinases of the parasite but also those of the host. A few cystatins have been molecularly characterized in the trematodes *Schistosoma* sp., *Fasciola* sp., and *Clonorchis sinensis*, for example, two *Fasciola gigantica* cystatins, both classified as type 1 cystatin, were found to be involved with regulating intracellular cysteine proteinase activity and protecting against extracellular proteolytic processes [8,9,10,11,12,13,14]. From our previous study of type 3 multi-domain cystatin in *Fasciola gigantica*, we found that processed cystatin domains inhibited both parasite and mammalian cysteine proteinases. We speculated that the processed domains were functional during spermiogenesis and fertilization. The major proteinases released by adult *Opisthorchis viverrini* are cysteine proteinases: cathepsins F and B. Both *Opisthorchis viverrini* cathepsins can digest human hemoglobin and extracellular matrix proteins, contributing to the development of CCA [15,16]. However, there are no laboratory data showing the functions and properties of *Opisthorchis viverrini* cysteine proteinase inhibitors. Through screening of transcriptome and genome databases of *Opisthorchis viverrini* [17,18], we found six transcripts/genes encoding cystatin type 1 and type 2 in adult *Opisthorchis viverrini*. Here, we have molecularly characterized these six *Opisthorchis viverrini* cystatins (*Ov*Cys1–6). Their sequence similarity is relatively low, except between *Ov*Cys1 and *Ov*Cys2. The native cystatins were found in the ES product of the adult stage, even though some of them contain no signal peptide. The inhibition properties were analyzed with mammalian cathepsins B and L, as was the stability at different pH values and high temperatures. This study contributes to our understanding of their biological functions. Other properties of *Opisthorchis viverrini* cystatins, such as anti-inflammatory effects, remain to be analyzed.

## 2. Materials and Methods

### 2.1. Parasites

Metacercariae of *O. viverrini* were collected from naturally infected fish. Female Syrian golden hamsters were infected with 100 metacercariae via gastric intubation. Adult *O. viverrini* were collected from the livers and bile ducts of 12-week-infected hamsters. The worms were washed twice in 0.85% normal saline solution and kept in liquid nitrogen until used in further experiments. The animal use protocol for all animal experiments in this study was approved by the Thammasat University Animals Ethics Committee (12 February 2020, Protocol No. 033/2561).

### 2.2. Molecular Cloning and Sequence Analysis

Total RNA was extracted from adult worms in TRIzol reagent (Invitrogen, Carlsbad, CA, USA) and used as a template for RT-PCR as described in our previous study [10,19]. Gene-specific primer pairs were designed from transcriptome/genome databases of *O. viverrini* [17,18], as detailed in Table 1. The PCR products were ligated into pGEM-T Easy (Promega, Madison, WI, USA) and the inserted sequences were determined by 1st BASE Sequencing Asia. EMBOSS (EMBOSS programs < EMBL-EBII) and SignalP 6.0 [20] were used to edit and analyze the sequences. MUSCLE version 5.2 https://www.drive5.com/muscle/ (accessed 29 May 2023) was used for multiple sequence alignment of *O. viverrini* cystatins (*Ov*Cys1–6) and human cystatin C (UniProtKB: P01034). The AlphaFold Protein Structure Database at “https://alphafold.ebi.ac.uk (accessed 14 July 2023)” was used to obtain models of the *O. viverrini* cystatins [21].

### 2.3. Expression of Recombinant Opisthorchis viverrini Cystatins and Production of rOvCys1–6 Antisera

The coding DNA sequences of six *Opisthorchis viverrini* cystatins (*Ov*Cys1–6) were subcloned into the expression plasmid pQE30 (QIAGEN, Hilden, Germany) using restriction sites introduced by the listed primers (Table 1). The recombinant plasmids were sequenced by 1st BASE Sequencing Asia. *Escherichia coli* M15 was used as an expression host, with induction by IPTG at 1 mM final concentration. Due to the insolubility of the expressed recombinant proteins, all *Opisthorchis viverrini* cystatins were purified under denaturing conditions using Ni-NTA affinity-chromatography, following the protocols in the QIAExpressionist manual (QIAGEN, Germany). Antisera against recombinant *Opisthorchis viverrini* cystatins were produced in ICR mice (two female mice per antigen) via the intraperitoneal cavity. Ten micrograms of each antigen were used to immunize mice three times in 3-week intervals. Pre-immune and immune sera were stored at −20 °C for further immunological experiments.

### 2.4. Parasite Antigen Preparation

Adult parasite-soluble crude worm extract (CWE) and excretory-secretory (ES) products were prepared as described [22]. Briefly, 200 adult *O. viverrini* worms were homogenized in lysis buffer using an UltraTurrax T25 tissue homogenizer (IKA, Staufen, Germany). The supernatant was collected as soluble CWE and used for further experiments. The ES product was prepared from fresh adult *O. viverrini*. A total of 500 worms were washed with normal saline solution and then incubated in 0.01 M PBS, pH 7.2, in 5% CO_2_ at 37 °C. The buffer was collected and centrifuged to eliminate insoluble material. The supernatant was concentrated using a centrifugal concentrator (3 kDa cut-off, GE Healthcare, Buckinghamshire, UK). Protein concentrations were measured using Bradford assay (Bio-Rad, Hercules, CA, USA).

### 2.5. SDS-PAGE and Western Blot Analysis

Twenty micrograms of each *O. viverrini* CWE and ES product and 100 ng of recombinant *O. viverrini* cystatins were size-separated using 16% SDS-PAGE and transferred to nitrocellulose membranes (Bio-Rad, Hercules, CA, USA) via semi-dry transfer using a Fastblot B33 instrument (Whatman, Biometra, Germany). Antisera specific to each recombinant cystatin and pre-immunized sera at a dilution of 1:1000 were used to probe the bound proteins. Alkaline phosphatase goat anti-mouse IgG (Sigma, Saint Louis, MO, USA) was used as a secondary antibody at a dilution of 1:30,000. The immune complex was enzymatically detected using BCIP/NBT phosphatase substrate (Amresco, Solon, OH, USA).

The cross-reactivity of each *O. viverrini* cystatin antiserum was investigated. To this effect, 100 ng of each recombinant cystatin was size-separated using SDS-PAGE, transferred to nitrocellulose membranes, and probed with antisera at the same dilution and via the same process described above.

### 2.6. Reactivity of Sera from Opisthorchis Viverrini-Infected Hamsters with Recombinant OvCys1, 3 and 4

*Ov*Cys1, 3 and 4 were tested with *O. viverrini-*infected hamster sera sampled 12 weeks post-infection (12 wpi, *n* = 10) and compared with normal hamster sera (0 weeks post-infection, 0 wpi, *n* = 10) using indirect ELISA. Recombinant H1_OvROPN1L was used as a positive control [23]. Briefly, microtiter plates were coated with 100 ng of each antigen in carbonate buffer. Skim milk at a concentration of 1% (*w*/*v*) was used to block non-specific binding. Individual hamster sera were diluted at 1:100 in TBS, pH 7.5, and 0.1% (*v*/*v*) Tween-20. Goat anti-(Syrian) hamster IgG AP-conjugated antibody (Abcam, Cambridge, MA, USA) was used as the secondary antibody at a dilution of 1:1000. p-Nitrophenyl phosphate (pNPP) substrate (Sigma Aldrich, St. Louis, MO, USA) was added and absorbance values were measured at 405 nm.

### 2.7. Protein Refolding, Inhibitory Activities of Recombinant OvCys1, 3 and 4 against Mammalian (Bovine) Cathepsin B and L, and Their pH and Temperature Stability

The recombinant *Ov*Cys1, 3 and 4 expressed in the bacterial host system were selected and their inhibitory activities tested against bovine cathepsin B and L. Due to the insoluble form of these expressed recombinants, they were refolded by step-wise dialysis in urea buffer as described [10] in order to obtain their soluble functional forms. The inhibition coefficients (IC_50_), pH and temperature stability of the refolded cystatins were determined as previously described [11,12]. The residual inhibitory activity of the cystatins (1 μM) was measured after incubating in buffer at different pH values (pH 3–9) for 30 min and at high temperature (99 °C) for 0–180 min. Soluble recombinant *Fg*Stefin2 was used as a positive control [12]. Each experiment was conducted in triplicate. The raw data were normalized between 0 and 100% activity and fitted nonlinearly based on a dose–response simulation (variable slope) to obtain the IC_50_ values using GraphPad Prism 9.5, GraphPad Software (LLC, Portland, OR, USA).

## 3. Results

### 3.1. Molecular Cloning and Sequence Analysis

Six cDNA sequences encoding cysteine proteinase inhibitors (cystatins) were isolated from adult *O. viverrini* via RT-PCR using specific primers (see Section 2.2). The nucleotide sequences were submitted to GenBank database under accession no. OR047397–402 (Table 2). The cDNA size, molecular weight and core motif of each cystatin are shown in Table 2. Cystatins are classified into three types: type 1 cystatins (single domain without a signal peptide and disulfide bonds), type 2 cystatins (single domain with a signal peptide and disulfide bonds) and type 3 (multidomain cystatins) [24]. Based on their sequences, *Ov*Cys1 and 2 were classified as type 1 cystatins and the other four cystatins, *Ov*Cys3–6, were classified as type 2 cystatins (Figure 1). The C-termini of *O. viverrini* cystatins show low conservation compared with human cystatin C. Following AlphaFold models, *Ov*Cys3–5 might have two disulfide bonds (Figure 1). These are in different positions compared with those of human cystatin C. *Ov*Cys6 lacks one of the four cysteines and thus has only a single disulfide bond. The amino acid sequences of *Ov*Cys1 and 2 share 84% similarity, while *Ov*Cys3–6 are less similar (30–65%) (Table 3). This high sequence conservation may be the cause of the cross-reactivity of *Ov*Cys1 and 2 antisera, as described in Section 3.3. Additionally, four of the five active site residues in human cystatin C are conserved. The core motif follows the consensus pattern QhV.G. (h hydrophobic, any residue) in all *O. viverrini* cystatins.

### 3.2. Expression and Purification of Recombinant O. viverrini Cystatins and Antisera Production

*O. viverrini* cystatins (*Ov*Cys1–6) were successfully expressed as insoluble proteins in *E. coli* M15. They were purified under denaturing conditions and used in denatured form for antibody production. Antisera against the six recombinant *O. viverrini* cystatins were raised in two mice per antigen. The antisera were used to detect native cystatins in the parasite soluble extract and ES product. Insoluble recombinant *Ov*Cys1, 3 and 4 were solubilized via step-wise dialysis and their soluble forms were used for inhibition assays of mammalian cathepsin B and L (Section 3.5).

### 3.3. SDS-PAGE and Western Blot Analysis

Soluble crude worm extract (CWE) and ES products were prepared from adult *O. viverrini*. The quality and quantity of parasite extracts were analyzed as mentioned in Section 2.3 and Section 2.5. Twenty micrograms of each *O. viverrini* CWE and ES product were size-separated on 16% SDS-PAGE and used in western blot detection (Figure 2a). Western blot detection showed that only mouse anti-r*Ov*Cys1, 3 and 4 antisera reacted with their immunizing antigens (r*Ov*Cys1, 3 and 4) in CWE and ES product, with monomer, dimer and trimer forms (Figure 2b). Anti-r*Ov*Cys2, 5 and 6 antisera reacted with the recombinant proteins but not with the parasite extracts. The test for cross-reactivity between antisera against each isoform revealed that anti-*Ov*Cys1 cross-reacted with r*Ov*Cys2, whereas anti-*Ov*Cys2 did not react with r*Ov*Cys1 (Figure 2b). This is likely due to the high sequence conservation between *Ov*Cys1 and 2 (Table 3). This cross-reactivity indicates that native *Ov*Cys2 may also be detected by anti-*Ov*Cys1 antisera.

### 3.4. Reactivity of O. viverrini-Infected Hamster Antisera with Recombinant OvCys1, 3 and 4

Individual sera from *O. viverrini*-infected (12 wpi) and uninfected (0 wpi) hamsters (*n* = 10, each) were used to probe r*Ov*Cys1, 3 and 4 and to analyze binding using indirect ELISA (Figure 3). The recombinant H1 peptide of *Ov*ROPN1L was used as a positive control [23]. The recorded absorbance values of the sera of uninfected and infected hamsters were not statistically different.

### 3.5. Inhibition of Mammalian (Bovine) Cathepsin B and L by Recombinant OvCys1, 3 and 4, and pH and Temperature Stability

Purified soluble r*Ov*Cys1, 3 and 4 were tested for inhibitory activity against bovine cathepsin B and L. *Fg*Stefin2 was used as a positive control [12]. *Ov*Cys1, 3 and 4 were selected to test inhibition because their native forms were found in soluble CWE and ES products (see Section 3.3). *Ov*Cys1, 3 and 4 showed similar inhibition of cathepsin B but were less efficient than *Fg*Stefin2 (Figure 4). The IC_50_ values showed highest inhibition activity for *Ov*Cys1, followed by *Ov*Cys3 and *Ov*Cys4, respectively (Table 4). The inhibition of cathepsin L was similar for *Ov*Cys1 and *Ov*Cys3 and only slightly lower than for *Fg*Stefin2. *Ov*Cys4 had slightly lower inhibitory activity, as observed for pH and temperature stability of two *F. gigantica* cystatins, *Fg*Stefin1 and 2 [11,12]. *Ov*Cys1, 3 and 4 also showed stability over a wide range of temperatures and pH values (Figure 5).

## 4. Discussion

In the present research six potential cystatins of the human liver fluke *Opisthorchis viverrini* were analyzed to determine their molecular sequence, protein distribution, antigenicity, and inhibitory function. These inhibitors of cysteine proteinases are commonly found in eukaryotes [24] but are also present in a smaller number of bacteria and viruses (InterPro: IPR000010). Flukes are thought to require cystatins to control the activity of endogenous cysteine proteinases and proteinases released by the host. In the case of *O. viverrini*, the parasite has been found to release cathepsin B, L and F—cysteine proteinases that are required for the digestion of host proteins—into the host environment [15,25,26]. The six investigated cystatins are single-domain cystatins and fall into two groups: type 1 cystatin without a signal peptide and disulfide bonds represented by *Ov*Cys1 and *Ov*Cys2 and type 2 cystatins that possess both signal peptide and disulfide bonds and are represented by *Ov*Cys3–6. The type 1 *Ov*Cys1 and *Ov*Cys2 are highly conserved, at 72% identity/84% similarity, whereas the type 2 *O. viverrini* cystatins show lower sequence conservation. Low sequence conservation is common among cystatins with only a few residues, mainly those associated with activity, being conserved.

While the six cystatin-specific antisera were all reactive with their cognate recombinant proteins, only antisera against r*Ov*Cys1, r*Ov*Cys3 and r*Ov*Cys4 detected the native proteins in CWE and ES products. Possibly, the other three cystatins were only present in small amounts in CWE and ES products and immunogenic detection was not sensitive enough to detect them. Refolded, soluble r*Ov*Cys1, r*Ov*Cys3 and r*Ov*Cys4 showed cystatin-typical high stability at high temperatures and across a broad pH range. They inhibited cathepsin L better than cathepsin B, which is normally explained by the excluding loop in cathepsin B that interferes with binding. High inhibition of cathepsin B was found for a *Fasciola gigantica* cystatin expressed in the parasite’s prostate gland and was later also demonstrated for the homologous cystatin in *F. hepatica* [12,27]. The recombinant *Ov*Cys3–6 used in this study contained the signal peptide; in the case of r*Ov*Cys3 and r*Ov*Cys4, this did not seem to affect their function.

In contrast to observations made for cystatins from the liver fluke *F. gigantica* [11,12], there was no immune response against these proteins in *O. viverrini-*infected hamsters. Of course, antigens released by *F. gigantica* in the juvenile intrahepatic stage stimulate a stronger immune response than antigens released by *O. viverrini*, a much smaller parasite that does not feed on blood, elicit in the bile ducts. However, other *O. viverrini* proteins do stimulate an immune response [28,29]. One explanation might be the low molecular weight of cystatin (11–13 kDa) and its complexing with cysteine proteinases that limit the number and accessibility of epitopes and, consequently, lead to low antigenicity. Based on the results, these cystatins are not applicable for the diagnosis of infection. Whether they are suitable drug targets remains to be determined; however, this is doubtful as each of them can bind to many different cysteine proteinases and thus knocking out one cystatin might have little impact.

In recent years, the roles of cystatins in various diseases, for example in cancer where they can interfere with invasion and metastasis, have been uncovered (reviewed in [30,31,32]) and have renewed interest in these inhibitors as research targets. Immunomodulatory activities of cystatins have been reported not only in Mammalia but also in Nematoda, Platyhelminthes [27,33] and Arthropoda [34]. Within the trematodes, cystatins have for example been identified in the digestive tract, prostate gland and tegument and are thought to be important for the regulation of endo- and exogenous cysteine proteinases [11,12].

## 5. Conclusions

Six single-domain cystatins of *Opisthorchis viverrini* were investigated. Three of them were found to be released in the excretion-secretion product of the parasite but did not stimulate a detectable immune response within the first 12 weeks of infection in hamsters. Activity assays confirmed their inhibitory properties as well as heat and pH stability. Further studies of these proteins should focus on their potential immunomodulatory activities.

## Figures and Tables

**Figure 1 pathogens-12-00949-f001:**
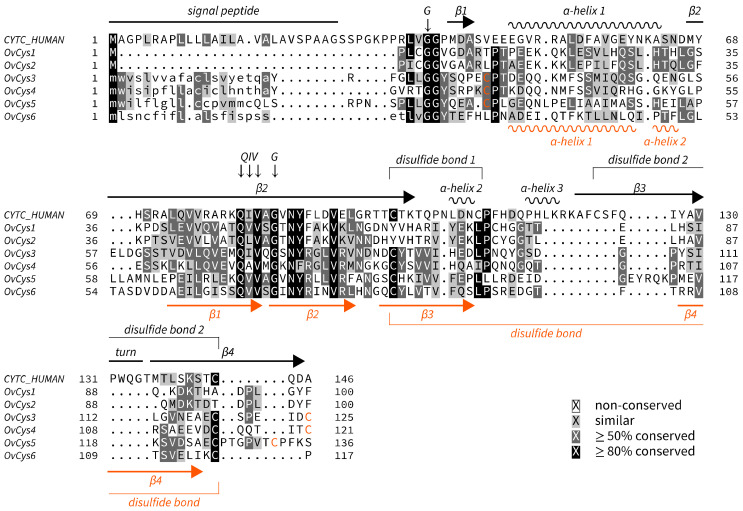
Multiple sequence alignment of *Ov*Cys1–6 and human cystatin C. Predicted *Ov*Cys3–6 signal peptides are shown in lowercase letters. The signal peptide, secondary structure and two disulfide bonds are indicated at the top, based on the reference human cystatin C (UniProtKB: P01034). Active site residues are indicated by downward arrows. The AlphaFold-predicted structure of *Ov*Cys3 (UniProtKB: A0A075AIV5) is shown at the bottom in orange color. Orange-colored cysteines in *O*vCys3–5 are predicted to form a second disulfide bond.

**Figure 2 pathogens-12-00949-f002:**
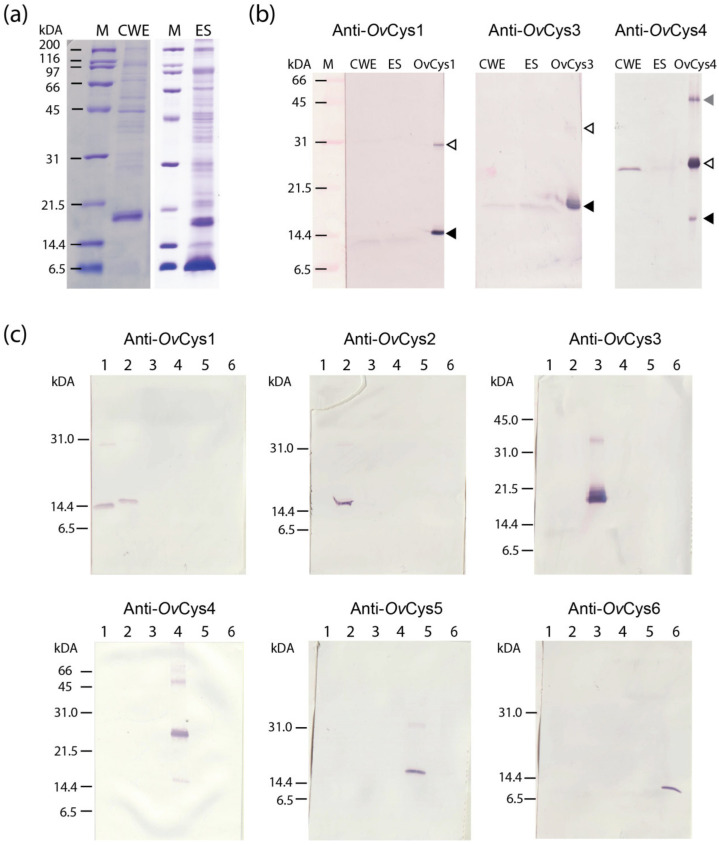
(**a**) SDS-PAGE of crude worm extract (CWE, 20 μg) and excretory-secretory (ES, 20 μg) products prepared from adult *O. viverrini*. (**b**) Western blot detection of CWE (20 μg), ES products (20 μg) and r*Ov*Cys1, 3 and 4 (100 ng each) reacted with mouse anti-*Ov*Cys1, 3 and 4 antisera. Black, white and grey arrows indicate monomer, dimer and trimer forms of the cystatins. (**c**) Cross-reactivity of anti-r*Ov*Cys antisera with each recombinant *O. viverrini* cystatin (100 ng each). Lanes 1–6 contain r*Ov*Cys1–6, respectively.

**Figure 3 pathogens-12-00949-f003:**
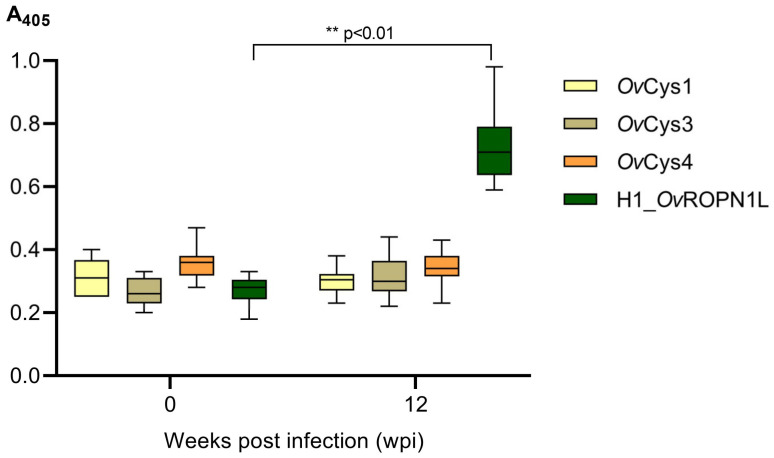
Indirect ELISA results obtained with sera from *O. viverrini*-infected (12 wpi) and uninfected (0 wpi) hamsters (*n* = 10, each). The whisker lines indicate the minimum and maximum values. The asterisk (**) represents statistical significance (*p* < 0.01) as calculated using the Wilcoxon matched-pairs signed-rank test.

**Figure 4 pathogens-12-00949-f004:**
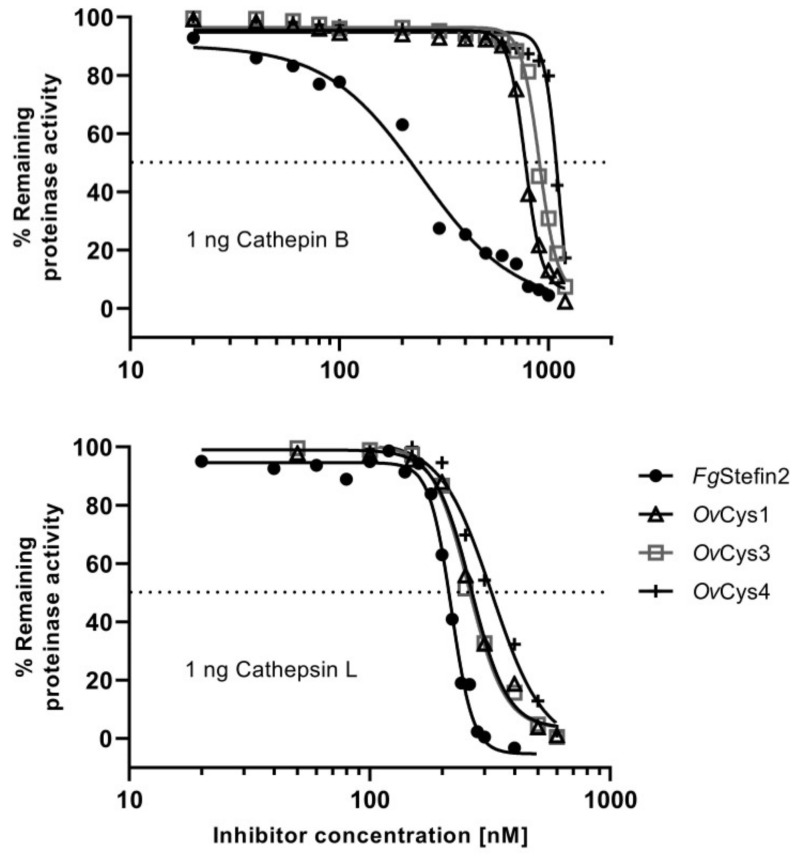
Inhibition of bovine cathepsin B and L by recombinant *Ov*Cys1, 3, and 4 in comparison with *Fg*Stefin2 [12]. The IC_50_ values are listed in Table 4.

**Figure 5 pathogens-12-00949-f005:**
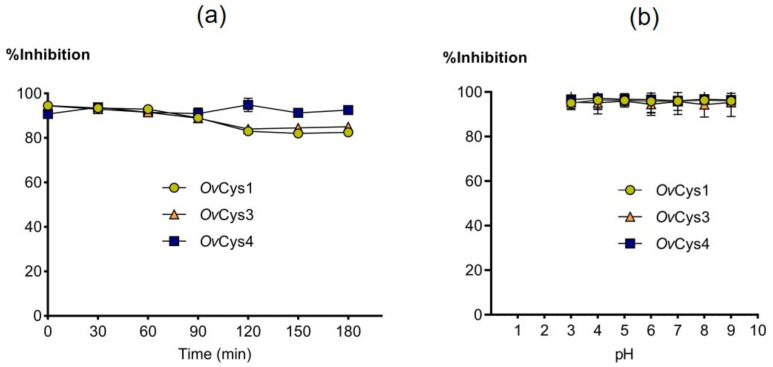
Graphs showing remaining activity of 1 μM of *Ov*Cys1, 3 and 4 against mammalian cathepsin L after incubation at 99 °C for 0–180 min (**a**) and in buffer of pH 3–9 for 30 min (**b**). Comparable stability has been reported for *Fg*Stefin1 and *Fg*Stefin2 [11,12].

**Table 1 pathogens-12-00949-t001:** Primer sequences with introduced restriction endonuclease sites (BamHI/PstI, underlined) for cloning of *O. viverrini* cystatins.

Isoforms	Primer Sequences	
	Forward (5′-3′)	Reverse (5′-3′)
*Ov*Cys1	GGATCCATGCCACTGTGCGGAGGT	CTGCAGTCAAAAATAGCCCAACGGGT
*Ov*Cys2	GGATCCATGCCAATATGCGGTGGCGT	CTGCAGTCAAAAATAATCCAACGGATC
*Ov*Cys3	GGATCCATGTGGGTCAGTTTGGTAG	CTGCAGTCAGCAGTCAATCTCTGGAC
*Ov*Cys4	GGATCCATTAAAGTCACCATGTGGA	CTGCAGTTAACATGTGATGGTTTGTTGAC
*Ov*Cys5	GGATCCATGTGGATTTTATTCCTG	CTGCAGTTAGCTTTTGAAGGGACAG
*Ov*Cys6	GGATCCATGCTGTCGAACTGCTTT	CTGCAGCTACGGACATTTTATGAGCT

**Table 2 pathogens-12-00949-t002:** Properties of *Opisthorchis viverrini* cystatins.

Isoform	GenBank Accession No.	cDNA Size (bp)	Amino Acids	Predicted MW (kDa)	Cystatin Core Motif (QVVAG)	Signal Sequence
*Ov*Cys1	OR047397	303	100	10.9	QVVSG	No
*Ov*Cys2	OR047398	303	100	11.1	QLVAG	No
*Ov*Cys3	OR047399	378	125	13.8	QIVQG	Yes
*Ov*Cys4	OR047400	366	121	13.4	QAVMG	Yes
*Ov*Cys5	OR047401	411	136	15.1	QVVAG	Yes
*Ov*Cys6	OR047402	354	117	12.9	QVVSG	Yes

**Table 3 pathogens-12-00949-t003:** Similarity (%) between amino acid sequences of cystatins from *Opisthorchis viverrini* (*Ov*) and *Fasciola gigantica* (*Fg*) Stefin2.

	*Ov*Cys1	*Ov*Cys2	*Ov*Cys3	*Ov*Cys4	*Ov*Cys5	*Ov*Cys6	*Fg*Stefin2
*Ov*Cys1		84	37.3	34.6	26.3	40	32.4
*Ov*Cys2			38.9	32.4	29.3	39.5	29.1
*Ov*Cys3				65.1	44.9	48.5	41.9
*Ov*Cys4					35.9	41.5	45.5
*Ov*Cys5						41.2	30
*Ov*Cys6							40.4
*Fg*Stefin2							

**Table 4 pathogens-12-00949-t004:** IC_50_ values of recombinant *Ov*Cys1, 3 and 4 and *Fg*Stefin2 against bovine cathepsins B and L.

**A**. **Enzyme: cathepsin B (1 ng/100 μL)**	Substrate: Z-Arg-Arg-AMC (10 μL)
Inhibitors:	IC50 (95% CI) (nM):
*Ov*Cys1	772 (754–793)
*Ov*Cys3	901 (869–957)
*Ov*Cys4	1104 (1047–?)
*Fg*Stefin2	246 (192–383)
**B**. **Enzyme: cathepsin L (1 ng/100 μL)**	**Substrate: Z-Arg-Arg-AMC (10 μL)**
Inhibitors:	IC50 (95% CI) (nM):
OvCys1	264 (247–288)
OvCys3	258 (243–277)
OvCys4	322 (288–404)
FgStefin2	218 (212–224)

## Data Availability

Nucleic acid sequences were submitted to GenBank database, as mentioned in the main text. Other data supporting the findings of this study are available from the corresponding author upon request.

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
