# Peer review of "Cystatins from the Human Liver Fluke *Opisthorchis viverrini*: Molecular Characterization and Functional Analysis"

_pathogens, 2023, doi:10.3390/pathogens12070949_

Round 1
Reviewer 1 Report
This work investigates 6 cysteine protease inhibitors from the Opisthorchis genome following on from the Fasciola work with similar genes. The work is solid and well presented. Minor edits are suggested in the attached pdf.

Author Response
Reviewer 1
Thank you very much for your valuable comments. We have responded to all as described below:
Page 1
Comment 1. The abstract has been revised as suggested.
Comments 2–5. Corrected as requested
Page 2
Comment 1: The primer does have an ATG, it is just further inside (GGATCCattaaagtcaccATGtgga).
Page 5
Comment 1: Yes, as mentioned in section 2.2. SignalP 6 was used to detect potential signal peptides. Figure 1 has been updated to show the predicted signal peptides in OvCys3–6. Furthermore, the inclusion of the signal peptides in the recombinant proteins has been mentioned in the discussion: "Recombinant OvCys3–6 used in this study still contained the signal peptide. In case of rOvCys3 and rOvCys4 this seemed not to have affected their function."
Comment 2: As suggested we have added structure predictions made by AlphaFold. AlphaFold models of OvCys3–5 suggest the presence of two disulfide bonds (Figure 1) in these three cystatins while OvCys6 would have a single disulfide bond. Of course that would need later experimental verification. We have updated section 3.1 to include this information: "According to their AlphaFold models, OvCys3–5 might have two disulfide bonds (Figure 1). These are in different positions compared to those of human cystatin C. OvCys6 is lacking one of the four involved cysteines and thus has only a single disulfide bond."
Page 8
Comment 1: edited
Comment 2 and 3: The details are already provided in Figures 4, 5 and Table 4 and should not be repeated in the text body.
Page 10
Comment 1: We have amended the legend of Figure 5: "Comparable stability has been reported for FgStefin1 and FgStefin2 [11,12]."
Reviewer 2 Report
The purpose of this manuscript is to introduce the molecular characterization and functional analysis of cystatins from Opisthorchis viverrini. This manuscript was described relatively well, and the contents would be informative to the journal readers. However, several major points should be addressed and revised properly in the manuscript.
Major comments
- The authors found six cystatin isoforms and then they name them OvCys1-6. But why didn't they name them as OvCystatins as shown in Table 1.
- OvCys1-2 and OvCys3-6 belong to type 1 and 2, respectively. Do type 1 and 2 indicate the official class or the authors' decision? To address cystatin phylogenetic membership clearly, several trematode cystatins should be included and analyzed. Based on the data, type 1 and 2 (or just clade 1 and 2, group 1 and 2) can be explainable.
- In the discussion section, various roles of cystatins in cancer (for example) were addressed. However, the final paragraph should suggest the potential roles of cystatins within the parasites in detail.
- Please address why the cystatins are important for parasitology research, for example, diagnostic molecules or drug targets. Which aspects do the authors focus on or else?
Minor comments
- line 80: SignalP 6.0 should be cited using the appropriate article, https://www.nature.com/articles/s41587-021-01156-3.
- lines 178-181: These sentences are redundant with the Materials and Methods section.
Typos
- line 7: @icloud -> @icloud.com
- line 18: OvCys2-6 -> OvCys3-6
- line 25: /F. gigantica/ -> /Fasciola gigantica/
- Abbreviations should be spelled out in full at first; and after the first appearance, the abbreviation should be used, such as CWE and ES
- In my opinion, the US can be omitted for materials, like line 109. But other materials came from the USA or the United States. Please keep your words consistent.
- line 149: Please provide complete information on GraphPad Prism 9.5, such as "(brand, city, state, country)".
The quality of the English language is quite good but there are several typos, such as species names and abbreviations.
Author Response
Reviewer 2
Thank you very much for your valuable comments. We have explained/edited/corrected as showed below:
Major comments
- The authors found six cystatin isoforms and then they name them OvCys1-6. But why didn't they name them as OvCystatins as shown in Table 1.
ANSWER: They are now consistently named OvCys throughout the manuscript.
- OvCys1-2 and OvCys3-6 belong to type 1 and 2, respectively. Do type 1 and 2 indicate the official class or the authors' decision? To address cystatin phylogenetic membership clearly, several trematode cystatins should be included and analyzed. Based on the data, type 1 and 2 (or just clade 1 and 2, group 1 and 2) can be explainable.
ANSWER:
Type 1, type 2 cystatins are official classifications as described in: Turk, V., Stoka, V., Turk, D., 2008. Cystatins: biochemical and structural properties, and medical relevance. Front. Biosci. 13, 5406-5420. Doi: 10.2741/3089
Replace reference 22 (too old) with the above publication.
- In the discussion section, various roles of cystatins in cancer (for example) were addressed. However, the final paragraph should suggest the potential roles of cystatins within the parasites in detail.
ANSWER: The discussion has been amended accordingly: “Within the trematodes cystatins have for example been located in the digestive tract, prostate gland, and tegument and are thought to be important for regulation of endo- and exogeneous cysteine proteinases [11,12].”
- Please address why the cystatins are important for parasitology research, for example, diagnostic molecules or drug targets. Which aspects do the authors focus on or else?
ANSWER: We have added two sentences to the discussion that together with the statements in the following paragraph should make clear that host modulation by these cystatins is of great interest to parasitologists. “Based on the results these cystatins are not applicable for diagnosis of infection. Whether they are suitable drug targets remains to be analyzed and doubtful as each of them can bind to many different cysteine proteinases and thus knockout of one cystatin might have little effect.”
Minor comments
- line 80: SignalP 6.0 should be cited using the appropriate article, https://www.nature.com/articles/s41587-021-01156-3. Corrected
- lines 178-181: These sentences are redundant with the Materials and Methods section. Section “Results 3.2” was edited.
Typos
- line 7: @icloud -> @icloud.com corrected
- line 18: OvCys2-6 -> OvCys3-6 corrected
- line 25: /F. gigantica/ -> /Fasciola gigantica/ corrected
- Abbreviations should be spelled out in full at first; and after the first appearance, the abbreviation should be used, such as CWE and ES Thank you Thank you very much for your suggestion
- In my opinion, the US can be omitted for materials, like line 109. But other materials came from the USA or the United States. Please keep your words consistent. corrected
- line 149: Please provide complete information on GraphPad Prism 9.5, such as "(brand, city, state, country)". corrected